Use of the traditional halibut hook of the Makah Tribe, the čibu.d, reduces bycatch in recreational halibut fisheries

Petersen Joseph R. 1 2
Scordino Jonathan J. jonathan.scordino@makah.com 1
Svec Cole I. 1
Buttram Reginald H. 1
Gonzalez Maria R. 1
Scordino Joe 3
1 Makah Fisheries Management, Makah Tribe , Neah Bay , WA , United States of America
2 Northwest Indian Fisheries Commission , Forks , WA , United States of America
3 Scordino Consulting , Edmonds , WA , United States of America
Montenegro Alvaro
Electronic publication date: 2020 Jun 12
Publication date: 2020
Volume: 8
Electronic Location ID: e9288
Received 2020 Feb 18; Accepted 2020 May 13
Copyright: ©2020 Petersen et al.
Copyright year: 2020
Copyright holder: Petersen et al.
License: This is an open access article distributed under the terms of the Creative Commons Attribution License, which permits unrestricted use, distribution, reproduction and adaptation in any medium and for any purpose provided that it is properly attributed. For attribution, the original author(s), title, publication source (PeerJ) and either DOI or URL of the article must be cited.
License URL: https://creativecommons.org/licenses/by/4.0/

Keywords: Pacific halibut, Traditional ecological knowledge, Hippoglossus stenolepis, Bycatch, Recreational fisheries, Fisheries management, Bycatch reduction, Makah Tribe, Traditional halibut hook, Hook design

Funding: National Marine Fisheries Service NA16NMF4270250 This project was funded by the National Marine Fisheries Service through a Saltonstall-Kennedy Grant (No. NA16NMF4270250). The funders had no role in study design, data collection and analysis, decision to publish, or preparation of the manuscript.

==============================
A previous study found that use of the traditional halibut hook (čibu.d) of the Makah Tribe in present day recreational Pacific halibut (Hippoglossus stenolepis) fisheries significantly reduced bycatch compared to paired 8/0 circle hooks. The study also found that the čibu.d had a significantly reduced catch of halibut, but that the reduction may have been due to manufacturing flaws in the čibu.d used in the study. In this two-phased study, we first compared the fishing performance of redesigned čibu.d made from four different materials: brass, stainless steel, plastic, and wood. In the second phase, we compared the fishing performance of the brass čibu.d with two common recreational fishing set-ups: a single large 16/0 circle hook and paired 8/0 circle hooks. The fishing performance of the redesigned čibu.d was not statistically different for čibu.d made of brass, stainless steel, or plastic. However, the čibu.d made from wood had significantly lower catch of halibut than the other čibu.d. We selected the brass čibu.d for the second phase of the study for continuity with the previous study of čibu.d and found that it had significantly less bycatch and a lower bycatch ratio than both the paired 8/0 and single 16/0 circle hooks. No significant differences were found in catch rates of halibut for paired 8/0 circle hooks, 16/0 circle hook, and the brass čibu.d. This study demonstrates that the improved catching performance of čibu.d on halibut and reduced bycatch compared to other popular approaches can be achieved by using brass čibu.d. Managers of recreational halibut fisheries should consider the use of čibu.d in areas where bycatch is a concern.

Introduction

The ecological impacts of recreational fishing have commonly been overlooked in favor of focusing on the impacts of commercial fisheries, but the effects of recreational fisheries can be significant to both the target species and bycatch (McPhee, Leadbitter & Skilleter, 2002; Coleman et al., 2004). Recreational fisheries can have higher impacts than commercial fisheries in localized areas (Cooke & Cowx, 2004). The effects of bycatch in recreational fisheries for Pacific halibut (Hippoglossus stenolepis; hereafter halibut) are poorly known due to lack of observer coverage and unknown amounts of bycatch discarded at sea (Cooke & Cowx, 2004; Lewison et al., 2004). Recreational halibut fisheries are monitored through port samplers checking retained catch and having anglers self-report their bycatch. Self-reporting may be accurate (Figus & Criddle, 2019) but it is very difficult to verify, making it difficult to evaluate just how much bycatch occurs in recreational halibut fishing. Bycatch released at sea is vulnerable to a variety of stressors, including risk of infection from hooking injuries, loss of predator avoidance, barotrauma, and other stress induced by time on deck, all of which can directly impact the ability of a bycaught fish to survive (Trumble, 1996). Recently, state management agencies have required the use of descending devices because recompressed rockfish have an improved probability of post-release survival (Hannah, Parker & Matteson, 2008; Hannah, Rankin & Blume, 2014; Bellquist et al., 2019). However, recent studies have found that rockfish may experience prolonged effects from barotrauma that can negatively affect their survival (Rankin et al., 2017). Thus some species bycaught in recreational halibut fisheries, because of their physiology (like rockfish) or because of their life history (like spiny dogfish) (Stevens et al., 2000), are vulnerable to overfishing even at modest levels of incidental mortality during fisheries. Improving the selectivity of fishing gear used in recreational halibut fisheries would make them more sustainable and minimize their impacts on non-target species.

The traditional halibut hooks used by Native Americans of the Pacific Northwest were designed and refined over thousands of years of trial and error to target halibut while minimizing catch of non-halibut species (Stewart, 1977). Prior to contact with Europeans, the halibut hooks were made from wood and had a bone barb and after contact metals were utilized in the construction first in making the barb and later in making the entire hook. Note that in this paper we use the term ‘barb’ to refer to the bone or steel that is added to the hook’s frame that stabs through a fish’s mouth to be consistent with past descriptions of traditional hooks (Swan, 1870; Stewart, 1977; Malindine, 2017; Salmen-Hartley, 2018). In a previous study, we evaluated if using the Makah Tribe’s traditional halibut hook (known as the čibu.d, Fig. 1) could reduce impacts of bycatch during recreational halibut fishing (Scordino et al., 2017). Our study confirmed that the čibu.d is more selective for recreationally catching halibut than modern hook configurations. However, the study also found that the čibu.d significantly reduced the catch of halibut when compared to commonly used paired 8/0 circle hooks (Scordino et al., 2017).

Figure 1 Side by side representation of the handmade čibu.d fished in the Scordino et al. (2017) study (A) and the improved čibu.d design used in this study (B).

Notable changes include: (1) Changes to the leader attachment method. (2) Standardization of the shape of the čibu.d frame and position of the barb 36 mm above the bottom of the čibu.d frame. (3) Changes in the material used for the barb. (4) Changes to methods to attach the barb for the improved čibu.d.

We identified two characteristics of the čibu.d fished in our previous study that affected its fishing performance (Scordino et al., 2017). Our previous study used handmade hooks that were variable in size and the position of the barb in the hook; we found that čibu.d that caught the most halibut had a mean distance of 36 mm in length from the tip of the barb to the bottom of the čibu.d frame (Fig. 1). This was a significantly larger gap than čibu.d that did not catch halibut (mean distance of 32 mm). Second, the straightened fish hook used for the barb of the čibu.d may also have been too weak, suggesting that a stronger barb would improve fishing performance (Scordino et al., 2017).

Although not evaluated in the previous study, the material used to manufacture the čibu.d may have also affected fishing performance. In 1880, fishermen of the Makah Tribe landed 719.5 metric tons of halibut while hand lining čibu.d (Collins, 1996). Traditionally, the čibu.d was made from steam bending a single piece of hemlock, true fir, or yew and affixing a barb made from bone or antler (Stewart, 1977). It is likely that many or most of the čibu.d used in 1880 were made of wood. The positive buoyancy of a wooden čibu.d may have made it more effective for catching halibut as compared to brass čibu.d that are negatively buoyant. This observation led us to hypothesize that fishing performance of the čibu.d tested in Scordino et al. (2017) may be improved by manufacturing the čibu.d from a more buoyant material such as wood or plastic.

Our previous study (Scordino et al., 2017) showed that the čibu.d was a promising tool for managers to use in recreational halibut fisheries with bycatch concerns. However, the significant reduction in observed halibut catch rates with the čibu.d likely would frustrate anglers who often assess fishing satisfaction based on catching their target fish (Arlinghaus, 2006). The objective of this study was to determine if the performance of the čibu.d for catching halibut could be improved through modifications of the čibu.d construction while maintaining the beneficial reductions in bycatch observed in our previous study (Scordino et al., 2017). To achieve this objective, we conducted a two-phase study. In phase 1, we evaluated if the material used to construct čibu.d affects their fishing performance by comparing catch rates on čibu.d made of brass, stainless steel, plastic, and wood. In phase 2 of the study, we compared the best performing čibu.d design from phase 1 of the study to two popular recreational halibut fishing approaches used today: fishing with paired 8/0 circle hooks and fishing with a single 16/0 circle hook. We evaluated fishing performance by comparing catch rates of halibut, bycatch rates, and bycatch ratios between the three methods.

Materials & Methods

Manufacture of čibu.d

Our study design required the manufacture of čibu.d from plastic, wood, stainless steel, and brass (Fig. 2). Each of the four material types required different manufacture protocols as described below. All improved čibu.d for this study were made with as similar a shape as possible to minimize the possibility that hook shape affected fishing performance. A documentary video on this project with a detailed demonstration of how the wooden čibu.d and brass čibu.d were handmade and how metal čibu.d were made on a compact metal bender is available at the Makah Museum in Neah Bay, WA.

Figure 2 Photograph of representative čibu.d used in phase 1 of this study and circle hooks used in phase 2 of this study.

(A) Wood čibu.d made of western hemlock branch with a bone barb made from elk femur and wraps of split Sitka spruce root. (B) Plastic čibu.d made of 33% glass filled nylon with a 316 stainless steel barb. (C) Brass čibu.d made of 360 half-hard tempered brass with a 316 stainless steel barb. This hook was used in phase 1 and 2 of this study. (D) Stainless steel čibu.d made of 316 stainless steel with a 316 stainless steel barb. (E) Paired 8/0 circle hooks. (F) 16/0 circle hook.

The design of the metal čibu.d used in this study were improved based on observations from our previous research (Scordino et al., 2017) in four ways (Fig. 1): (1) we used 0.3175 diameter 316 stainless steel rod for the barbs of the čibu.d instead of straightened fish hooks; (2) we welded or silver soldered the barb to the frame of stainless steel and brass čibu.d, respectively, rather than securing the barb by wrapping in wire and soldering with plumbing solder; (3) we standardized the shape of the čibu.d frame and position of the barb inside the frame with the barb tip positioned 36 mm above the bottom of the frame; and (4) we drilled a hole through the top of the čibu.d frame to secure a leader rather than securing a large barrel swivel directly to the čibu.d frame. The revised attachment of the leader made the improved čibu.d smoother on the inside of the frame than the previously tested čibu.d.

The frames of the brass and the stainless steel čibu.d were made from rods of 360 half-hard tempered brass and 316 stainless steel with a diameter of 0.635 cm and a length of 30.5 cm. We used a bench grinder to taper the rod down to a one mm diameter tip on one side of the rod starting three cm from the end. We achieved a consistent taper by making a tool for the bench grinder to guide the angle at which the rods were tapered. For quality control, we compared each tapered rod to a reference rod with an ideal taper and discarded any that were incorrectly tapered. We developed a stepwise procedure to shape the metal čibu.d frames with high precision and consistency using a compact metal bender. All čibu.d frames were shaped by author RB to further ensure consistency of shape. The barbs for both the brass čibu.d and the stainless steel čibu.d were made out of 0.3175 cm diameter 316 stainless steel rod that was cut to five cm and tapered to a sharp point. The barb was welded onto the stainless steel čibu.d frame and silver soldered onto the brass čibu.d frame with the barb tip positioned 36 mm above the bottom of the frame. We wrapped the bottom of the frame with cotton twine to add texture to hold the bait in position. At the balancing point of the top of the čibu.d frame on both the brass and stainless steel čibu.d a two mm hole was drilled horizontally for the attachment of a fishing wire leader and barrel swivel.

The wooden čibu.d were made with traditional materials used by the Makah Tribe as generally described by Stewart (1977). Materials collected to construct the wooden čibu.d were western hemlock (Tsuga heterophylla) branches, Sitka spruce (Picea sitchensis) root, tallow, and elk femur. Sections of western hemlock branches with a diameter of about four cm and length of 30.5 cm were radially cut into sixths to make blanks for shaping into čibu.d. The blanks were shaved to achieve a triangular shaped cross-section by removing wood on the pith side and preserving the continuous grain along the bark side to maintain the strength of the finished čibu.d. The blank was tapered to a narrow tip starting three cm from one end of the blank. The completed hemlock blank was steamed until pliable and bent and secured around a form with the tapered end oriented on the top of the čibu.d frame and the pith side of the blank was aligned towards the inside of the form. Once affixed to the form, we cured the blank at room temperature for 24 h. After curing, the blank was notched with a slot to attach the barb. We then heated the frame and applied tallow to the entire blank to seal the wood against moisture and help it hold shape (Stewart, 1977). A sharpened piece of worked elk femur of roughly 10 cm was affixed in the notch using thin strips of spruce root so that the tip of the barb was roughly 36 mm from the bottom of the frame of the čibu.d. The spruce root wrap secured the barb and also provided texture to hold the bait in position by continuing wrapping the frame from the barb insertion point to a position directly below where the where the leader was secured to the top of the čibu.d frame. The tip of the čibu.d frame opposite where the barb was attached (top of frame) was also wrapped in spruce root to mimic how the Makah traditionally made the hook (Fig. 2). All of the wooden čibu.d were handmade and although we attempted to standardize the shape and position of the barb in each, there was variability in the completed čibu.d.

We contracted Benchmark Molding of Edmonds, Washington to make the plastic čibu.d. Benchmark Molding created a design similar in shape to the metal čibu.d used in this study. The shape of the plastic čibu.d was different in two ways. First, two bumps were added near the balancing point of the hook on the top of the frame for tying a leader in place. Second, the bottom of the čibu.d was designed with ribs wrapping around the čibu.d to help hold the bait in place, thus negating the need to wrap the čibu.d in cotton twine. Like the metal čibu.d, we used a five cm rod of 0.3175 cm diameter 316 stainless steel rod that was tapered to a sharp point. The barb was securely held in place by being encapsulated within the plastic of the čibu.d frame and held with the barb tip positioned with a gap from the point of the barb to the bottom of the frame of the čibu.d of 36 mm. A 33% glass filled black nylon was injected to make the semi-rigid frame of the čibu.d. Tag ends from the injection process were trimmed to leave the final shape of the čibu.d.

Field deployment

We conducted our field tests using nearly identical methods to our previous study (Scordino et al., 2017). We contracted Windsong Charters for both phases of the study to provide the vessel and fishing equipment sans the terminal gear used in the study. Volunteers were recruited from the local community to be the anglers for the study. During both phases, fishing took place off the coast of northern Washington, with all sites accessible from the port of Neah Bay (Fig. 3). The test hooks were fished off a 50.8 × 20.3 cm ‘L’ shaped spreader bar commonly used in recreational fishing that had a 0.9-kilogram weight attached to the short side and the hook attached to the long side. We baited all hooks with brined, blue label herring (18 –20 cm) to avoid bait size causing a bias in our study (Kaimmer, 2004).

Figure 3 Map showing sites fished during the two phases of the study.

Fishing sites are shown by symbol for the years they were fished: circles were fished in 2017 only, squares were fished in 2018 only, and triangles were sites fished in both years. The star shows the location of the Port of Neah Bay from which research was conducted.

The sampling unit for each phase of the study was a 30-minute set. Prior to each set, the poles were set up with the hook type alternating down the rail of the boat; two of the same hook type were never fishing next to one another at the start of a set. We instructed anglers to actively fish through the entire set and to catch as many fish as possible. When anglers caught fish, we recorded species, length, and the hook type that caught the fish. When anglers hooked a fish, or reeled up to check bait, we quickly rebaited their hooks to fish as continuously as possible during a set. At the conclusion of each set, we rotated anglers counterclockwise to a new position on the boat and a new hook type. Rotating the anglers to new positions on the boat reduced the potential for bias caused by fishing location on the vessel and ensured that anglers fished all hook types. The rotation of anglers also eliminated the possibility of angler skill affecting catch rate. Our goal was to complete at least six sets per sampling day but the actual number of sets per day was variable due to factors including weather, distance from port to sampling locations, and energy of the volunteers to continue fishing with consistent effort.

Our study design was very similar for both phases of the study except for a few notable differences. First, in phase 1 we fished two replicates of each of four čibu.d types (brass, stainless steel, plastic and wood) with eight anglers whereas in phase 2 we fished three replicates of three hook arrangements (brass čibu.d, paired 8/0 circle hooks, and 16/0 circle hook) with nine anglers. Second, our deployment of hooks from the spreader bar was different during the two phases of study. During phase 1, we attached the čibu.d to the spreader bar using leaders of 6 to 10 cm in length for all of the čibu.d. During phase 2, the spreader bar was attached to either a brass čibu.d with a 6 to 10 cm leader, a single 16/0 circle hook with an 8 to 12 cm leader, or paired 8/0 circle hooks with 12 to 16 cm leader to the top hook and gap between the top and bottom hook of around six cm. The leaders used for each hook type differed in length in order to minimize tangling during deployment. At the conclusion of each day of fishing during phase 1 we had anglers rank their preference for the čibu.d by material type from most to least preferred.

During phase 1 of this study, 75 experimental sets were conducted during 13 days of effort in June (29–30) and July (5–7, 12–14, 17–19, and 27–28) of 2017 in the Pacific Ocean and Strait of Juan de Fuca in the waters surrounding northwest Washington (Fig. 3). Sets ranged in depth from 85 to 207 m with an average set depth of 149 m. A bottom discrimination function on a Standard Horizon sonar was used to record substrate type. Fifty-eight of the sets were conducted over sand or gravel bottom substrate and an additional 17 sets were conducted in substrate mixed with rock or primarily rocky bottom. Phase 2 was conducted over 36 experimental sets over 5 days of fishing from June 20–28, 2018 at some of the same sites used in Phase 1 and at additional sites where bycatch rates are generally higher. Twenty-four of the sets were conducted in rock/gravel substrate and 12 were sand/gravel substrate.

The International Pacific Halibut Commission provided permits for the research activities performed in 2017 (permit # EL2017070) and 2018 (permit # EL2018045). For all species other than halibut, impacts from this research were recorded against the treaty set-asides within section 50 CFR 660.50.

Data analysis

An ANOVA was used to analyze catch rates by hook type for halibut, bycatch species in aggregate, and for each species of fish caught for each phase of this study. We used a Tukey Honestly Significantly Different (HSD) test to determine what pair-wise comparisons of hook types caused the observed significant differences in ANOVA tests.

The bycatch ratios by hook type was evaluated with all catch data pooled during phase 2 of the study. A X2 test of independence was used to compare the ratio of halibut caught to bycatch. We conducted post hoc pairwise comparisons of bycatch ratios using a Bonferroni correction for multiple comparisons.

To evaluate if our modifications to the čibu.d outlined above improved the fishing performance of the čibu.d, we compared our results to our previous study (Scordino et al., 2017). We fished paired 8/0 hooks as a reference for evaluating the fishing performance of čibu.d both during phase 2 of this study and in our previous study. To test if our design improvements to the čibu.d improved fishing performance for catching halibut we compared the ratio of halibut caught on čibu.d to halibut caught on paired 8/0 circle hooks in our previous study (Scordino et al., 2017) to results from phase 2 of this study with a Fisher’s Exact Test. We also tested if the improved čibu.d maintained its beneficial reductions of bycatch relative to paired 8/0 circle hooks by comparing the ratio of bycatch caught on the čibu.d and the paired 8/0 hooks from our previous study (Scordino et al., 2017) to results from phase 2 of this study using the improved čibu.d.

Results

Phase 1

A total of 286 halibut were caught with 82 on brass, 81 on plastic, 82 on stainless steel and 27 on wood čibu.d. In addition to Pacific halibut, we also caught seven individual fish as bycatch. There were two bycaught on brass čibu.d, three on plastic čibu.d, one on stainless steel čibu.d, and one that was entangled in the angler’s fishing line and not actually hooked. Bycaught fish were of the species petrale sole (Eopsetta jordani), lingcod (Ophiodon elongatus), spiny dogfish (Squalus acanthias), and greenstripe rockfish (Sebastes elongatus).

We found strong evidence of differences in halibut catch rate by čibu.d type (ANOVA, p < 0.001, F = 10.6, df = 307). The observed differences in catch rates were due to significantly less catch of halibut on wood than on stainless steel, brass, or plastic čibu.d (Tukey HSD, p < 0.001 for all comparisons to wood; Fig. 4). No evidence was found for differences in catch between the brass, stainless steel, and plastic čibu.d.

Figure 4 Halibut caught per set by čibu.d made of each of the four materials tested.

Comparison of rate of fish hooked and lost

We found strong evidence of differences in rates of landing a fish once hooked by hook type (Fisher’s Exact Test, p < 0.001). Anglers landed (brought aboard the boat) 62% of fish hooked on brass čibu.d, 60% on stainless steel, 53% on plastic and 19% on wooden čibu.d. No evidence was found for statistical differences in whether or not a hooked fish was landed between the plastic, brass, and stainless steel čibu.d (Fisher’s Exact Test, p = 0.66).

Angler surveys

We conducted post-fishing interviews with 69 of the 77 volunteers. We found strong evidence that anglers preferred which type of čibu.d they fished (Fig. 4, Friedman test, X2 = 13.62, df = 3, p = 0.004). A post hoc test revealed that the preference was driven by anglers having a significantly stronger preference for brass, stainless steel or plastic than for wood (p = 0.020 and p = 0.007, respectively). No difference was found in angler preference for plastic, brass, or stainless steel čibu.d.

Phase 2

During phase 2 of the study we compared fishing performance of brass čibu.d to a single 16/0 circle and to paired 8/0 circle hooks. Our choice of the brass čibu.d was primarily to have continuity with our previous study (Scordino et al., 2017). Other factors that influenced our decision were that the brass čibu.d was easier to manufacture than the stainless steel čibu.d and it had similar performance and angler preference as the stainless steel and plastic čibu.d.

Anglers caught 346 fish consisting of 205 Pacific halibut and 141 fish of bycatch species during phase 2 of the study (Table 1). A total of 1.4 halibut were caught on paired 8/0 circle hooks for every 1 caught on čibu.d and 1.2 halibut on a single 16/0 circle hook for every 1 caught on a čibu.d. The halibut catch rate was not statistically different by hook type (ANOVA, df = 2, 102, p = 0.49; Fig. 5).

Table 1 Average catch per set by species and species groups by hook type during phase 2 of the study.

Species or species groups with significant differences (p < 0.05) in catch rate by hook type are marked with an asterisk.

	Average catch per set	
Species	Paired 8/0 circle hooks	16/0 circle hook	čibu.d	
Pacific halibut	2.257	1.971	1.629	
Bycatch species pooled*	2.571	1.200	0.257	
Non-halibut flatfishes pooled*	0.514	0.257	0.086	
Petrale sole	0.229	0.114	–	
Arrowtooth flounder	0.286	0.143	0.086	
Roundfishes pooled*	2.057	0.943	0.171	
Rockfishes pooled	0.457	0.200	0.086	
Canary rockfish	0.057	0.057	–	
Redstripe rockfish	0.029	0.029	–	
Rosethorn rockfish	0.029	–	–	
Tiger rockfish	0.029	–	–	
Yelloweye rockfish	0.286	0.114	0.086	
Yellowtail rockfish	0.029	–	–	
Pacific spiny dogfish*	0.743	0.571	0.029	
Lingcod*	0.571	–	0.057	
Sablefish	0.257	0.171	–	
Coho salmon	0.029	–	–	

Figure 5 Catch of halibut and bycatch per set for paired 8/0 circle hooks, a single 16/0 circle hook, and a brass čibu.d fished during phase 2 of the study.

(*) In this study we defined all catch of species other than Pacific halibut as bycatch.

There was strong evidence for differences in bycatch by hook type (ANOVA, df = 2, 102, p < 0.0001; Fig. 5). The paired 8/0 circle hooks had roughly ten times more bycatch than the čibu.d, and the 16/0 had roughly five times more bycatch than the čibu.d (Fig. 5). A Tukey HSD test revealed strong evidence for difference in bycatch on čibu.d and paired 8/0 circle hooks (p < 0.0001), a single 16/0 circle hook and paired 8/0 circle hooks (p = 0.0026), and suggestive evidence of difference in catch of bycatch on a čibu.d and a 16/0 circle hook (p = 0.054).

Likewise, our analysis using the pooled data from all phase 2 sets showed strong evidence for differences in bycatch ratios (halibut caught per bycatch species caught) by hook type (X2 = 31.43, df = 2, p < 0.0001). The bycatch ratios (halibut:bycatch) for the three hook types were 0.88:1 for paired 8/0 circles hooks, 1.6:1 for the single 16/0 circle hook, and 6.3:1 for the brass čibu.d. A post hoc pairwise comparison revealed evidence of differences in the observed bycatch ratios between paired 8/0 circle hooks and the 16/0 circle hook (p = 0.049) and strong evidence for differences in bycatch ratios for čibu.d and the paired 8/0 circle hooks (p < 0.0001) and the čibu.d and the single 16/0 circle hook (p = 0.003).

Significant differences in catch rates per set were found for all flatfish pooled and all roundfish pooled on different hook types (Table 1). None of the species of flatfish independently had significant differences in catch rates by hook type. The differences in catch of lingcod and spiny dogfish appear to have driven the significant differences in catch rates for roundfish as both independently had strong evidence of significant differences in catch rate by hook type.

Evaluating performance of improved čibu.d as compared to previous study

The ratio of one halibut caught on our improved čibu.d to every 1.4 halibut caught on paired 8/0 circle hooks in phase 2 of this study was significantly better than our previous čibu.d design that had 1 halibut caught on čibu.d for every 2.9 caught on paired circle hooks (Fisher’s Exact Test, p = 0.001). The ratio of bycatch caught on čibu.d to paired 8/0 circle hooks was not significantly different between the two studies (Fisher’s Exact Test, p = 0.12).

Discussion

The objective of this study was to evaluate if design and manufacturing modifications to the čibu.d tested in our previous study (Scordino et al., 2017) could improve halibut catch rates while maintaining reductions of bycatch relative to contemporarily fished circle hooks. To achieve this objective, we conducted a two-phase study. In phase 1 of the study, we evaluated čibu.d manufactured of four different materials that were thought to have different buoyancy and thus fish differently. We found that plastic, brass, and stainless steel čibu.d all caught halibut at a similar rate suggesting that the buoyancy and weight of the čibu.d did not affect its fishing performance for halibut. In phase 2 of the study, we compared the brass čibu.d to two contemporary hook designs commonly fished by recreational halibut anglers: a single 16/0 circle hook and paired 8/0 circle hooks. We found no statistical differences in the catch rate of halibut on the čibu.d, the paired 8/0 circle hooks, and the single 16/0 circle hook. The improved brass čibu.d still drastically reduced bycatch; the čibu.d caught four times more halibut per bycaught fish than the single 16/0 circle hook and seven times more halibut per bycatch fish than the paired 8/0 circle hooks. Compared to our previous design (Scordino et al., 2017), the improved brass čibu.d used in this study had nearly a twofold better catch rate of halibut relative to concurrently fished paired 8/0 circle hooks.

Our finding that the brass, stainless steel, and plastic čibu.d outperformed the wooden čibu.d during phase 1 of this study should not be viewed as a definitive finding that the materials make better čibu.d. The wood čibu.d were the most difficult to make and as a result we had much more variability in their shape and barb position than we had in čibu.d made with the other materials. It is quite likely that some of our wood čibu.d performed poorly due to their shape alone as some did not have the optimum shape we determined in our previous study (Scordino et al., 2017). We also found that some of our wood čibu.d either did not hold shape when fished due to saturating with water or were otherwise very weak resulting in an almost three fold reduction in catch retention as compared to the čibu.d made from other materials. The variability in the structural strength of our wood čibu.d was likely because we made wood čibu.d using both tension wood and compression wood not knowing until the study was under way that tension wood has less structural strength (Arima, 1975). In this study, we tried to mimic wooden hooks made by master builders who learned from their parents and grandparents who had previously mastered the craft. If our wooden čibu.d were made by master artisans as wood čibu.d were in the past, then it is likely that the catch rates of wooden čibu.d would have been more similar to the other materials. Likewise, it is quite possible that our metal replicates of Makah halibut hooks did not perform as well as metal čibu.d used in the past by tribal members who fished the hooks regularly and modified their hooks to optimize performance.

We do not know what makes the čibu.d selective for catching halibut. Stewart (1977) speculated that only flatfish, whose mouths are oriented perpendicular to the orientation of roundfish could successfully slide their mouth between the barb and the frame of the čibu.d. Bycatch on čibu.d in this study may have occurred when the orientation of the čibu.d was shifted when anglers bounced their weight along the bottom making it so roundfish could more easily pass their mouth between the barb and frame of the čibu.d. We encourage researchers to film halibut, yelloweye rockfish, spiny dogfish, and lingcod in either the wild or in a controlled aquarium setting to observe the mechanics of how each attacks a baited čibu.d to determine what attributes of the čibu.d shape influences whether or not the fish is hooked. Understanding what attributes reduce bycatch may allow modifications to the čibu.d design, or to the design of other hooks, to further improve catch selectivity.

Although the objective of this study was to evaluate the fishing performance of čibu.d, the study design also allowed us to compare the fishing performance of two hook configurations commonly used by recreational anglers. Many halibut anglers in Washington selectively use larger hooks to reduce bycatch of non-target species (Scordino et al., 2017). Our results showing a nearly twofold reduction in bycatch per set and twofold increase in halibut caught per bycatch caught on the single 16/0 circle hook as compared to the paired 8/0 circle hooks indicate that anglers using a larger circle hook have made the correct choice to reduce bycatch.

Bycatch in fisheries, even recreational fisheries, can be detrimental to the recovery of depleted fish stocks (Cooke & Cowx, 2004; Hall & Mainprize, 2005; Ihde et al., 2011). The results of this study demonstrate that hook selection significantly affects bycatch in recreational halibut fisheries. Recreational anglers can be educated on the benefits of selecting large (>16/0) circle hooks to encourage their voluntary use to reduce bycatch during halibut fishing (Cooke et al., 2013). Likewise, resource managers can mandate the use of large circle hooks in management areas with bycatch concerns. Currently the čibu.d is not commercially produced and available to the public. Once the čibu.d is available to the public, voluntary use by anglers, or regulations that require its use, would have even more conservation benefit in recreational halibut fisheries than the use of large circle hooks.

Conclusions

In our previous study (Scordino et al., 2017), we found that čibu.d could potentially be used as a tool by managers to reduce bycatch in recreational halibut fisheries. The caveat to that conclusion was that recreational anglers would likely not voluntarily switch to using the čibu.d due to the observed significant reductions in halibut catch rates compared to contemporary hook designs (Arlinghaus, 2006). In this study we improved the design of the čibu.d and showed that if a čibu.d is properly designed and constructed that it has similar catch rates of halibut as commonly fished circle hooks while maintaining beneficial reductions of bycatch. After modification, the čibu.d is now a tool that recreational anglers and fisheries managers can use to reduce bycatch in recreational halibut fisheries while not sacrificing halibut fishing performance. We strongly encourage the commercial manufacture of čibu.d to allow the use of this selective hook in recreational halibut fisheries. We encourage requiring the use of large hooks (16/0 or larger) in areas where bycatch is a concern in recreational halibut fisheries and requiring the use of čibu.d when they are commercially available.

We would like to thank the deckhands and captains of the Windsong particularly B Maxson, D Dawson, and M Hunter. Makah Tribal member, J Monette, played a critical role in research and development of the wood čibu.d used in this study. Thank you to the Makah Cultural and Research Center for input on making wood čibu.d and allowing us to examine historic samples. The manuscript was reviewed and edited by E Allyn. Thank you to the Makah Tribal Council and the International Pacific Halibut Commission for supporting this study. We would also like to thank the 119 volunteers that participated as anglers during the study. Views and opinions expressed in this paper are the authors and do not necessarily represent the Makah Tribe or the National Marine Fisheries Service.

Additional Information and Declarations

Competing Interests

Author Contributions

Field Study Permissions

Data Availability

Joseph R. Petersen works for the Northwest Indian Fisheries Commission and was a former employee of the Makah Tribe; Jonathan J. Scordino, Cole I. Svec, Reginald H. Buttram, and Maria R. Gonzales are employed by Makah Fisheries Management; and Joe Scordino is a retired employee of the National Marine Fisheries Service who runs his own consulting firm, Scordino Consulting.

Joseph R. Petersen and Jonathan J. Scordino conceived and designed the experiments, performed the experiments, analyzed the data, prepared figures and/or tables, authored or reviewed drafts of the paper, co-wrote the grant application and co-administered the budget, and approved the final draft.

Cole I. Svec performed the experiments, prepared figures and/or tables, and approved the final draft.

Reginald H. Buttram and Maria R. Gonzalez conceived and designed the experiments, performed the experiments, prepared figures and/or tables, and approved the final draft.

Joe Scordino performed the experiments, authored or reviewed drafts of the paper, and approved the final draft.

The following information was supplied relating to field study approvals (i.e., approving body and any reference numbers):

Field experiments were approved by the International Pacific Halibut Commission for research on Pacific halibut for 2017 (permit # EL2017070) and 2018 (permit # EL2018045). Research approval and approval to count non-halibut catch against tribal catch quotas was given by Makah Fisheries Management Director Russell Svec on behalf of the Makah Tribal Council. A National Environmental Policy Act analysis was completed by National Marine Fisheries Service for all Pacific Fishery Management Council managed species. It was deemed appropriate that all non-halibut impacted within this study were directly recorded against the Treaty set asides and allocations as set forth in section 50 CFR 660.50.

The following information was supplied regarding data availability:

Data compiled and analyzed during this study are available at Mendeley Data: Scordino, Jonathan (2020), “Testing of traditional Makah halibut hooks”, Mendeley Data, v1. http://dx.doi.org/10.17632/g8b298crnp.1.

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
