# Peer review of "Use of the traditional halibut hook of the Makah Tribe, the čibu.d, reduces bycatch in recreational halibut fisheries"

_PeerJ, doi:10.7717/peerj.9288_

## Round 0.1 · original submission · Minor Revisions

Thank you for your very interesting submission. I would like to second and add to the recommendation of Reviewer 3 regarding more discussion (be it in the intro or discussion sections) on the physical and functionality differences between circle and čibu·d hooks. Maybe a picture with the circle hooks could be helpful.

Also, it appears earlier stages of the project started out testing the hypothesis - which experiments did not confirm - that wooden čibu·d hooks would work better due to their buoyancy. I believe the manuscript would be enriched if the authors could provide a hypothesis, even if speculative, for the recorded reduction in by-catch.

·

Basic reporting

Well written work. See note in general section below on the term "barb". To avoid confusion with commonly accepted usage of barb (i.e barbed vs barbless hook rules in recreational fisheries, particularly recreational salmon fisheries) you may wish to use čibu·d point or čibu·d tip or "hook" point or "hook tip".

Experimental design

No comment.

Validity of the findings

Lines 335 through 341 make a great shout out to ancillary data results that support 16/0s over paired 8/0s as another method to reduce bycatch.

Additional comments

Line 38. Replace “environmental” with “ecological” (species changes caused by fish are more ecological than environmental. Environmental would be more accurately attributed to gear interactions with the environment).
Line 40. Remove word “ecologically” as it is now dealt with above.
Line 41. Add “Pacific” in front of northwest.
Paragraph bounded by lines 60-66 should reference Figure 1.
Also, there seems to be a nuance throughout the paper with hook part definitions that should be cleared up. The author appears to be using the term “barb” when really meaning the hook tip or hook point. A barb is generally viewed as the small “flange” off the point (see example on Figure 1a – left image) vs no barb in the image in Figure 1b – right image. In fact, some recreational fisheries require “barbless” hooks. In that case a barb is required to be removed or bent back flush with the main body of the hook point, they don’t require a hook without a hook point or hook tip. Because “barbed and barbless hooks” are a common vernacular in recreational fisheries, I would recommend that the author change all uses of “barb” to hook point or hook tip (global replace), and also possibly include a figure with the parts of the hook labelled (point, shank, gap, etc. similar to conventions used by Stewart, 1977 or in this web sourced diagram of the anatomy of a hook: https://www.bing.com/images/search?view=detailV2&ccid=REikvACz&id=99C77B30419BD8C63E062F4C54628C51BB54E90E&thid=OIP.REikvACzS9daGSE30hQUQwHaFC&mediaurl=https%3a%2f%2famateuranglers.files.wordpress.com%2f2015%2f09%2fparts-of-a-hook.jpg&exph=500&expw=736&q=parts+of+a+fish+hook&simid=608029633287028876&selectedIndex=9&ajaxhist=0


Line 73. Buoyancy in this sense is understandable, drag not so much. I would suggest removing the word “drag”.
Line 76. “more buoyant material” possibly add “such as plastic” to the end of that sentence.
Line 79. I would recommend “with rockfish bycatch concerns” rather than simply “with bycatch” concerns. The study investigates all bycatch concerns, but is driven by rockfish bycatch concerns, and oral history of specific čibu·d for halibut vs red fish.
Line 105-106. Design improvements were applied to all four different materials…except #2 the welding/soldering step was not done on plastic čibu·d. Later on the description of how the wooden čibu·d bone point is attached, but no mention of the how the point was anchored in the nylon/plastic čibu·d.
Line 111. Add “Neah Bay, WA”, to Makah Museum, or footnote details to where it is located. Not all readers will know.
Line 134. How was the tapering achieved? Each by hand, or was it machined so all were still the same length and same sharpness? Please describe.
Line 141 mentions twining the metal to increase bait retention to čibu·d. Line 149 mentions ribbing on the plastic čibu·d for the same purpose. Maybe be clear how that is achieved for the wooden variety as well.
Line 219-middle 224. Effort and location data should be moved to your methods section. Describe how the bottom substrates mentioned in lines 223 and 224 were obtained.
Line 259-262. See previous note, in that these would be more appropriate in the methods section. Also, how do these locations differ if at all from those discussed for Phase 1? If different goals for different sites, list those.
Line 284. Reads “catch rates per set calwere found” change to “catch rates per set were found”.[remove ‘cal’].
Line 308. “The improved brass čibu·d used in this study had nearly a twofold better catch rate of halibut relative to concurrently fished paired 8/0 circle hooks as compared to our previous design (Scordino et al., 2017).” While true, this sentence feels misleading, and it is easy for the reader to surmise that the čibu·d fishes twice as well as the paired 8/0’s. I recommend first describing how the hook compares to the 8/0s in the current study, before clarifying how much improved it is over the first study. Alternatively, the sentence could be rearranged to be a little less misleading, such as:
Compared to our previous design (Scordino et al., 2017) the improved brass čibu·d used in this study had nearly a twofold better catch rate of halibut relative to concurrently fished paired 8/0 circle hooks”.
I still think a discussion of catch rates of the improved čibu·d in relation to recreational standards of paired 8/0’s and single 16/0’s alone should come before this discussion, as even the suggested reword comes off sounding like the čibu·d outfishes the 8/0s, which is not what you are trying to say (for eg. Line 291/2.

Line 324. “…otherwise very weak.”. Maybe reference here the results section describing halibut retention of the other čibu·d types (from lines 238-243).
Line 346. “selecting the use of large” change to “selecting large”. [strike ‘the use of’]
Lines 372, 374. Avoid starting a sentence with a single initial (e.g. M.Hunter.J.Monette ). Makes distinction between ending of one sentence and the beginning of the next fuzzy. Rearrange the sentence structure to avoid.

Figure 1. Use a scale reference so reader knows how large the čibu·d are. Consider modifying or adding an figure with hook parts labelled, including the 36mm gap location.

Figure 5. Use a footnote number beside the term bycatch to define that it includes all non-halibut. Then, add an asterisk to the image if any of those differences are significant, which I believe lines 268-282 backup. List the stats in the title.

Table 1. Show significance, as able.

Reviewer 2 ·

Basic reporting

Please see "General comments for the author" below.

Experimental design

Please see "General comments for the author" below.

Validity of the findings

Please see "General comments for the author" below.

Additional comments

Summary: In this study, the authors tested the efficacy of traditional Makah fishing hooks (cibud) fabricated using multiple materials for catching the target species—halibut—while reducing bycatch. The study built on a previous experiment that showed the cibud significantly reduced bycatch but also reduced halibut catch relative to the most commonly used recreational fishing hook. In that previous study, there may have been hook design flaws that resulted in reduced halibut catch. Here, the authors first compared the performance of four cibud hook types made from different materials (plastic, brass, stainless steel, and wood; Part 1). They then compared the best performing cibud from Part 1 with two common methods of recreational fishing for halibut (2 8/0 circle hooks and 1 16/0 circle hook; Part 2). The plastic, brass, and stainless steel cibud hooks caught similar numbers of halibut as the 8/0 and 16/0 circle hooks while significantly reducing bycatch of other species. The traditional hook may be a superior alternative to commonly used recreational fishing gear.

I found this to be a straightforward study with clear, informative results that have important implications for recreational halibut management and conservation. The paper was well written and easy to follow, with a strong study design and robust statistical methods. My suggested edits are minor and I recommend acceptance following revision.

Substantive comments--
My only bigger picture comment is that with some limited additional language in the Introduction and Discussion sections the authors could more clearly illustrate and better communicate the broader impacts of their study to increase its chances of being seen by the broader community of halibut researchers, fishers, and managers. I recommend adding some brief language in both the Introduction and Discussion (2-4 sentences per section) regarding bycatch concerns and knowledge gaps in halibut fisheries. Which bycatch species are of greatest concern (e.g., rockfishes)? What work has been done on bycatch in sport halibut fisheries (if any)? Although this study is focused on recreational fishing, I also recommend citing bycatch studies related to commercial halibut fisheries to show that there has been a great deal of attention on the bycatch issue (e.g., Figus and Criddle 2019, https://www.sciencedirect.com/science/article/pii/S0308597X1730667X).

It also strikes me that the results could be relevant to management of the subsistence halibut program in Alaska (SHARC). People typically fish for subsistence halibut using skates with 30 circle hooks and bycatch is of some concern, although there is limited documentation. This might be worth a mention in the Discussion.

Finally, in the Discussion, I suggest that you comment on the relative costs (materials, labor) involved in fabrication of various hook types. If someone were to invest in manufacturing such hooks, what type might yield the best cost effectiveness?

Minor comments--
Line 51. Consider changing “Indian tribes” to “Indigenous peoples” as the latter term may be more widely recognized in other areas in the range of Pacific halibut, such as Alaska.

Line 64. Please specify the mean size of the hooks that did not catch halibut. This could be done parenthetically at the end of this sentence.

Lines 88-89. “two recreational fishing approaches used today” – please indicate where they are used. I believe the methods are also used commonly in Alaska, but it would be helpful to confirm. Are they also used in British Columbia? If similar gears are used to catch Pacific halibut throughout their range, the results of this study have relevance to a broader geographic extent than Washington alone.

Line 111. It’s great to hear there is a video available at the Makah Museum. Please include the address and website URL (if available) in parentheses at the end of this sentence so that readers have more information about the location.

Paragraph beginning Line 166. I noted that you reported the number of trips in the results, which is fine. However, here it would be helpful to report the mean number of sets done per trip.

Lines 178-180. Specify which hook types were used in each phase to avoid any confusion by the reader.

Line 187. Replace “form” with “from”

Line 226. Seven fish seems very low, especially compared to the Phase 2 experiments; is this number correct? Or do you mean seven species?

Line 240. This sentence initially confused me as written. I think clearer wording is: “Anglers landed (i.e., brought aboard the boat) 62% of the fish hooked on brass čibu·d…”

Lines 265-266. Include a post-hoc test to show which pairs of hooks were significantly different from each other.

Lines 272-273. Revise “suggestive evidence of” with “a weakly significant”, which is more common wording to describe such a result in fisheries and ecological studies.

Lines 279-280. It appears that a word is missing from this sentence.

Line 284. Replace “calwere” with “were”

Lines 308-310. This sentence is somewhat confusing as written; please revise to make clearer.

Figure 2. I suggest include a photo of the commercially made 8/0 and 16/0 circle hooks that were used in the study for comparison. Not all readers will be familiar with their design.

Figure 4. Indicate significant differences among hook types using lowercase letters or symbols on the figure (based on the post hoc test results).

Figure 5. What does the asterisk on this figure refer to? Also, need to indicate in the figure caption what the error bars represent.

·

Basic reporting

The paper is well-written, clear and unambiguous. The objectives and hypotheses are clearly stated, with relevant results. There are only 14 references; the authors could perhaps refer to and discuss other studies where circle hooks have been compared with other models of hooks, with the objective of reducing by by-catch of fish and turtles (e.g. pelagic longlines).

Experimental design

The paper presents the results of a follow-up study addressing the issue of by-catch mitigation in a particular recreational fishery. This is an important issue and the study is well designed, with adequate sample sizes and numbers of anglers.
My only question is why is there no information on sizes of the halibut caught with the different hooks? Size of the fish caught is important for angler satisfaction and it would be important to know if the traditional hook catches the same size range as the circle hooks. If it catches smaller fish, then this could be a problem for angler acceptance.

Validity of the findings

The findings are clear and conclusive. The research is original and the results useful for management and conservation.

Additional comments

If possible, please provide information on size distributions of the catches with the different hooks, along with results of a statistical test (e.g. Kolmogorov-Smirnoff test for comparing distributions).
As an keen angler myself, I am interested in the catch mechanism of the traditional hook. It might be interesting to include some text on this and perhaps a photo showing the way the traditional hook works (presumably rather like a circle hook?).

---

## Round 0.2 · accepted · Accept

The authors did a very good job in addressing reviewer concerns, which wore mostly minor to start with.